# Photopolymerized Films with ZnO and Doped ZnO Particles Used as Efficient Photocatalysts in Malachite Green Dye Decomposition

**Viorica-Elena Podasca \***  **and Mariana-Dana Damaceanu**

Petru Poni Institute of Macromolecular Chemistry, 41 A Grigore Ghica Voda Alley, 700487 Iasi, Romania; damaceanu@icmpp.ro

\* Correspondence: podasca.viorica@icmpp.ro

**Abstract:** Zinc oxide and zinc oxide doped with tin oxide ($ZnO-SnO_2$) particles were synthesized and successfully incorporated into a polymeric matrix by the photopolymerization reaction in the presence of Irg819 as the photoinitiator. The obtained samples were investigated by means of XRD, ESEM/EDX, TEM, FTIR, and Raman spectroscopy. The ZnO particles were obtained in the form of rods agglomerated in flower (or star) structures with lengths of 2–4 μm and widths between 30 and 100 nm, while $ZnO-SnO_2$ samples evolved in the form of cubes, with sides of 350 nm. The prepared composite films with ZnO and $ZnO-SnO_2$ particles were tested in the photocatalytic degradation of malachite green (MG) dye. While the ZnO-based composite film showed a fairly high photocatalytic activity, the hybrid film containing ZnO doped with $SnO_2$ displayed 100% photocatalytic activity after only 45 min of irradiation, being among the most efficient photocatalysts known for MG degradation. In addition, the recycling tests demonstrated that this film displayed high stability during the photocatalysis reaction since no decrease in the photocatalytic performance was noticed after the first three cycles, indicating its suitability for dyes removal and wastewater purification.

**Keywords:** ZnO; $ZnO-SnO_2$; photopolymerization; composite; malachite green; photocatalysis

## 1. Introduction

The increasing amount of pigments and dyes existing in the wastewater coming from the dye manufacturing and textile finishing industry significantly affects the water quality and reduces the photosynthetic reaction of aquatic flora by preventing the sunlight to reach it. Also, some of these dyes are known to be toxic with carcinogenic effects [1,2]. The dyes that accidentally arrive in the wastewater are very difficult to destroy using regular purification methods due to their high stability to oxidizing agents, light, and even to aerobic digestion [3,4]. In recent years, various processes based on both chemical and physical principles have been used to treat the wastewater effluents containing different dyes, among which electrochemical methods, hypochlorite, or ozone oxidation could be mentioned. These methods are usually costly, non-damaging, and ineffective since secondary waste products may result in these processes [5]. Therefore, it is necessary to find new tools and methods for removing dyes from the water.

Recently, fairly extensive research was concentrated on the development of new semiconducting materials for different applications, namely, photocatalysts for water disinfection and cleaning [6–11]. Over time, many photocatalysts, such as $TiO_2$, ZnO, $SnO_2$, CdS, etc., have been applied for the decomposition of organic dyes and air pollutants [12–14]. Among them, $TiO_2$-based materials have proved to have a very good catalytic activity due to their appropriate properties and large specific surface area that allow reactions to occur [15,16]. This is the reason why $TiO_2$ is the most suitable

semiconductor used in catalytic applications, such as the degradation of organic pollutants or oil spills from wastewater [17,18]. Apart from $TiO_2$, the second well-known photocatalyst is ZnO, having a significant role in materials research due to its high stability, low price, and nontoxicity. However, both $TiO_2$ and ZnO semiconductors have some limitations, such as agglomeration of the smaller particles into significantly larger ones, poor response to visible-light, or fast recombination rate of the photogenerated holes and electrons [19–21]. These issues lead to a decreased photocatalytic efficiency that obstructs the potential practical applications of these photocatalysts in the degradation of contaminants in air and water. Several methods have been applied to overcome these problems; in the case of $TiO_2$, for instance, the photocatalysts can be dispersed in several clay minerals, including montmorillonite, saponite, or nanotubes, to solve the agglomeration problems [22,23]. To increase the photocatalytic activity of ZnO, different approaches, such as reductive reactions [24,25], coupling with metal oxides [26] or semiconductors, such as $LaFeO_3$ [27], metal or non-metal doping, e.g., Mg, Ni, Sn [28–30], or surface modification [31] have been developed. The photocatalytic performance of ZnO could also be enhanced by modifying the morphology through different synthetic methods, like electrochemical deposition [32], electrospinning [33], vapor-phase synthesis [34], they have been used for the synthesis of ZnO materials with specific nanostructures [35,36].

In our previous studies [37–39], the development of nanosized ZnO with flower and spherical shapes was reported, as well as ZnO coupled with Ag nanoparticles that were obtained in situ during the photopolymerization process. Using this method, a mixture of monomers and nanoparticles was used to achieve flexible films whose photocatalytic activities were evaluated by using Nile red, methylene blue, and rhodamine B as model organic dyes. According to literature data, an improvement of the photocatalytic activity may be obtained by doping ZnO with metals [40–42]. Considering this, our current research was focused on the development of ZnO doped with tin dioxide ($SnO_2$), another inorganic material broadly used in photocatalysis due to its photosensitivity, transparency, and chemical stability. $SnO_2$ has excellent properties, such as chemical and thermal stabilities, and found applications in dye-sensitized solar cells [43], photocatalyst [44], gas sensors [45], and so on [46–48]. In this study, the doping of ZnO particles with Sn ions was carried out by a chemical process, and their physicochemical characteristics were thoroughly explored; the purpose of this doping was to make structural modifications and the possibility to use these particles in simulated solar light photodegradation experiments. Besides, a new polymer/ZnO-$SnO_2$ hybrid composite material was prepared through a convenient combination of particles and photopolymerization process, designed for efficient photodegradation of malachite green dye.

## 2. Results and Discussion

### 2.1. Structural Characterization of the Monomer

The N,N-(diisopropylcarbamoyloxy)ethyl methacrylate (N-MA) monomer was obtained by the reaction of N,N-diisopropylcarbamoyl chloride with 2-hydroxyethyl methacrylate (HEMA), its structure being fully confirmed by FTIR, [1]H, and [13]C NMR analyses. The [1]H and [13]C NMR spectra of N-MA monomer are illustrated in Figure 1a,b, respectively. The [1]H NMR spectra highlighted the presence of characteristic unsaturated monomer protons at 6.15 and 5.58 ppm due to the trans/cis configuration of methylene protons at 4.37-4.26 ppm, methine protons at 3.85 ppm, and methyl protons at 1.21 ppm. Supplementary confirmation of the correct monomer structure was brought by [13]C NMR analysis. In Figure 1b, the peaks at 166.16 and 154.38 ppm are assigned to the ester carbon, while the peaks at 135.08 ppm and 124.88 ppm are attributed to the carbon atoms engaged in the formation of the double bond. The carbon atoms from the methylene group were found at 61.96 ppm and 59.68 ppm; the ones from the methine group arose in the interval between 46.41 and 44.87 ppm, while the methyl carbons appeared at 19.57 ppm and 17.35 ppm.

In the FTIR spectrum shown in Figure S1, Supporting Information (SI), the monomer N-MA displayed absorption bands characteristic for C-H stretching vibrations at 2971-2877 $cm^{-1}$, carbonyl

groups at 1723 cm$^{-1}$, C=C bonds at 1638 and 815 cm$^{-1}$, and C-O-C unit at 1291 cm$^{-1}$. This analysis brought additional proof for the right structure of the synthesized monomer.

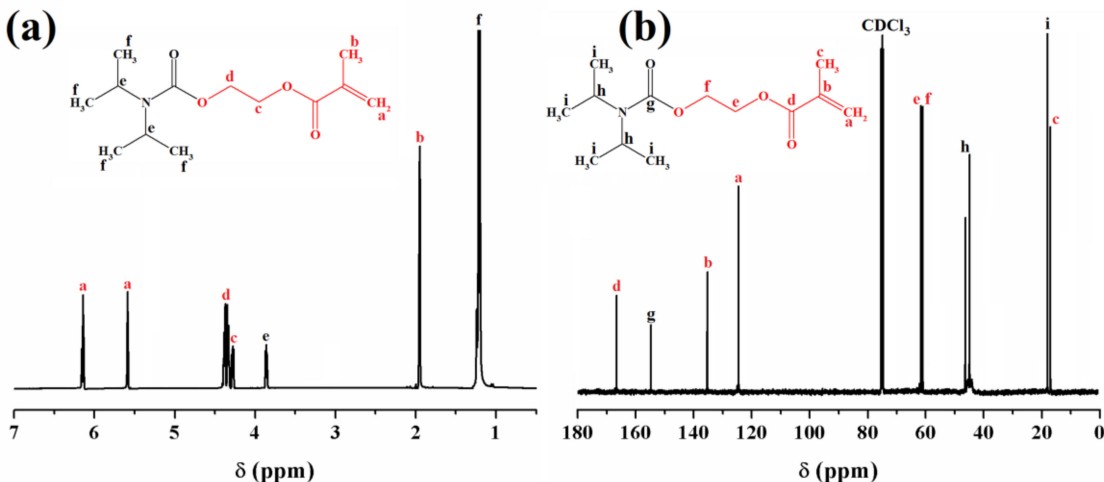

**Figure 1.** $^{1}$H NMR (**a**) and $^{13}$C NMR; (**b**) spectra of N,N-(diisopropylcarbamoyloxy)ethyl methacrylate (N-MA) in CDCl$_3$.

## 2.2. Particle Characterization

One of the main objectives of this study was the development and investigation of some hybrid composites containing ZnO-doped particles intended for photochemical purification of wastewater. Besides the synthesis of the monomer, ZnO and ZnO-SnO$_2$ particles were prepared by the method detailed in the Experimental section. Figure 2a shows typical X-ray diffraction (XRD) patterns for the obtained ZnO and ZnO-SnO$_2$. The pure ZnO sample displayed distinct peaks at 2θ = 31.8°, 34.4°, 36.2°, 47.6°, 56.6°, 62.9°, 68°, 69.2°, 66.4°, 72.6°, 77.1° corresponding to (100), (002), (101), (102), (110), (103), (200), (112), (201), (004), and (202) planes, with no evidence for any secondary phase or possible impurity, which was in line with the standard hexagonal (wurtzite) ZnO powder (standard JCPDS card no. 80-0075) [49]. The XRD spectrum of ZnO-SnO$_2$ exhibited broad diffraction peaks at 26.1°, 34.3°, and 52.2°, which could be attributed to the (110), (101), and (211) diffraction planes characteristic for the tetragonal structure of SnO$_2$. The characteristic peaks of ZnO could not be accurately distinguished with respect to those of SnO$_2$. Thus, it could be assumed that the ZnO-SnO$_2$ sample was in an amorphous phase [50–52].

Figure 2b illustrates the Raman spectra of the ZnO and ZnO-SnO$_2$ particles at room temperature. The peaks recorded for the ZnO sample at 101, 277, 331, 379, 438, and 580 cm$^{-1}$ could be associated with the hexagonal ZnO structure, the spectrum being almost identical to the reference one, reported in the literature [53,54]. On the other hand, significant peak shifts could be observed for the ZnO–SnO$_2$ sample as compared to the pure ZnO, changes that are probably caused by the appearance of heterostructures between the ZnO and SnO$_2$. For this sample, the peaks were found at 71, 404, 667, and 580 cm$^{-1}$, the last one characteristic for ZnO being in the form of a small shoulder [47,48]. The peak at 667 cm$^{-1}$ could be associated with the oxygen vacancies of SnO$_2$ nanocrystal surface defects [55,56].

The FTIR analysis of undoped ZnO and ZnO-SnO$_2$ particle samples was also performed (Figure S2a, SI). The results revealed that the peaks registered at 3435 cm$^{-1}$ and 1637 cm$^{-1}$ for ZnO and 3400 cm$^{-1}$ and 1639 cm$^{-1}$ for ZnO-SnO$_2$ originated from the adsorbed H$_2$O molecules on the particles surfaces. The bands at 516 and 393 cm$^{-1}$ could be associated with the absorptions of the Zn-O bond [57]. The incorporation of Sn to ZnO nanostructures led to the appearance of a large band, much intense compared to that of the pure sample at 560 cm$^{-1}$, which could be attributed, according to the literature data, to the Sn–O bond vibration [58].

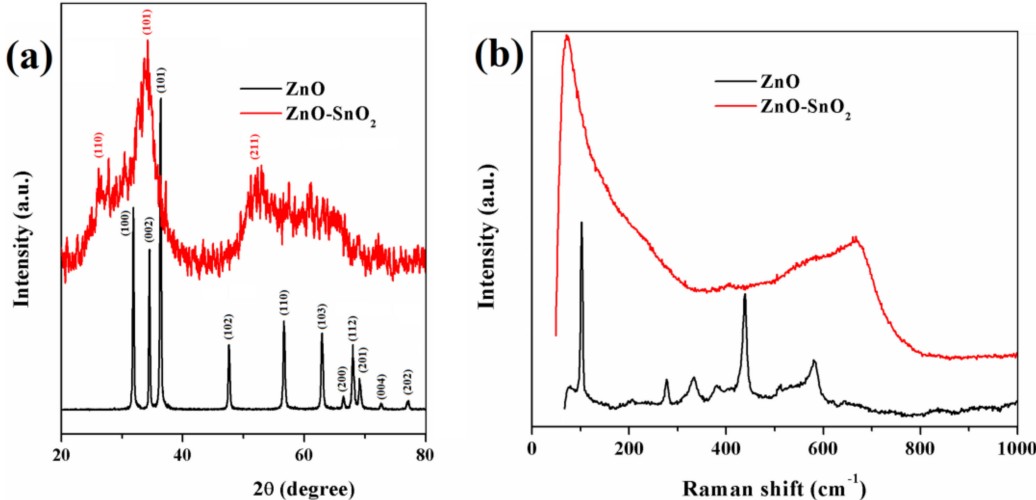

**Figure 2.** X-ray diffraction patterns (**a**) and Raman spectra (**b**) of ZnO and ZnO-SnO$_2$ powders.

Figure 3 illustrates the SEM and TEM micrographs of ZnO and ZnO-SnO$_2$ particles. The SEM images (Figure 3a) showed that the ZnO sample was in the form of flowers (or stars) with homogeneous rod shape. TEM analysis (Figure 3b) supported this observation, demonstrating that ZnO particles were in the form of homogeneous rods crowded in floral geometries with lengths of 2–4 μm and widths between 30 and 100 nm. For ZnO-SnO$_2$ particles, the SEM images (Figure 3c) evidenced their form of cubes with sides of approximately 350 nm, as calculated from the TEM analysis (Figure 3d). Compared to the shape of ZnO nanoparticles doped with SnO$_2$, reported in the literature [7,29,30], which have irregular or spherical shapes, our particles displayed well-defined cubic forms, never seen before.

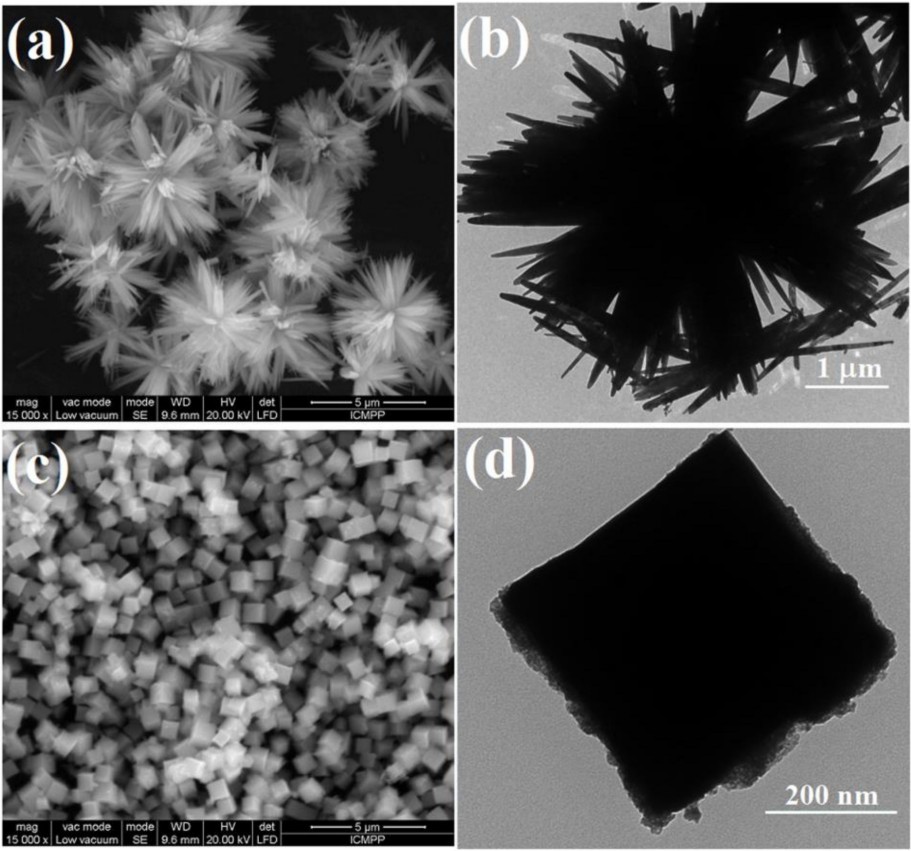

**Figure 3.** SEM and TEM images of ZnO (**a**,**b**) and ZnO-SnO$_2$ (**c**,**d**).

The elemental and spatial distribution of the tin and zinc atoms within the ZnO-SnO$_2$ composite was also explored by using the energy-dispersion X-ray (EDX) spectroscopy. According to Figure 4, the EDX chart clearly evidenced that Sn, Zn, and O atoms were the main components of the sample. The observed peak characteristic for C atoms is supposed to come from the instrument holder. The mapping images indicated that Zn and Sn were uniformly distributed in the analyzed sample.

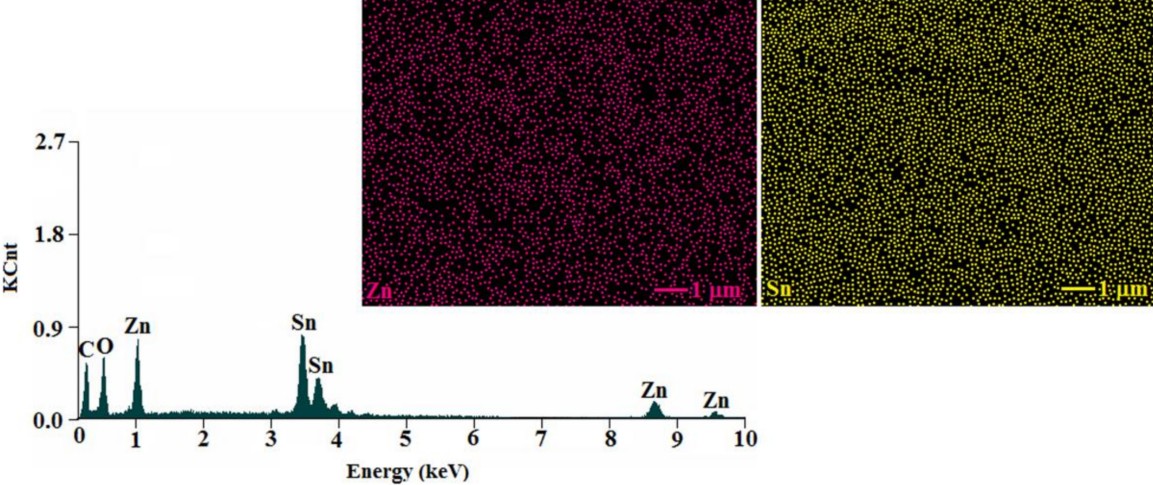

**Figure 4.** The spectrum of the cube-like ZnO–SnO$_2$ nanoparticles and mapping images for zinc and tin.

### 2.3. Synthesis and Characterization of Polymer and Polymer Hybrid Films

Polymer films were prepared by the photopolymerization of poly(propylene glycol) dimethacrylate (PO-UDMA) and N,N-(diisopropylcarbamoyloxy)ethyl methacrylate (N-MA) in a percent weight ratio of 60:40, by using 1.5 wt.% Irg819 as photoinitiator, at room temperature. The same method was applied for the synthesis of hybrid polymer films when the monomers were photopolymerized in the presence of ZnO or ZnO-SnO$_2$ particles (Ps) (Scheme 1), according to the formulations presented in Table 1. The irradiation was carried out with a 365 nm UV lamp at a light intensity of 40 mW/cm$^2$.

**Scheme 1.** Synthesis path towards ZnO or ZnO-SnO$_2$ samples.

**Table 1.** Data for **L1**, **L2,** and **L3** formulations.

| Sample | Composition (wt.%) | | | | Conversion Degree (%) |
|--------|--------|------|-----|---------|--------|
| | *PO-UDMA* | *N-MA* | *ZnO* | *ZnO-SnO$_2$* | |
| L1 | 60 | 40 | - | - | 98.15 |
| L2 | 60 | 40 | 1 | - | 92.94 |
| L3 | 60 | 40 | - | 1 | 93.10 |

The conversion degree of the characteristic methacrylate double bond in the obtained formulations was assessed on the basis of FTIR spectroscopy; the related data is presented in Table 1. As could be seen in Figure 5, the C=C double bond stretching vibration at 1638 cm$^{-1}$ fell with the enhancement of the UV exposure time, the conversion degree being approx. 98.15% after 60 s of irradiation in the case of sample L1. Under the same regime of lightning, the methacrylate double bonds (C=C) of samples L2 and L3 were transformed into C-C bonds with a conversion degree of 92.94% and 93.10%, respectively [59]. These results were better compared to our previous reports in which the PO-UDMA monomer was used in combination with another synthesized monomer [38] or with a commercially available one [39].

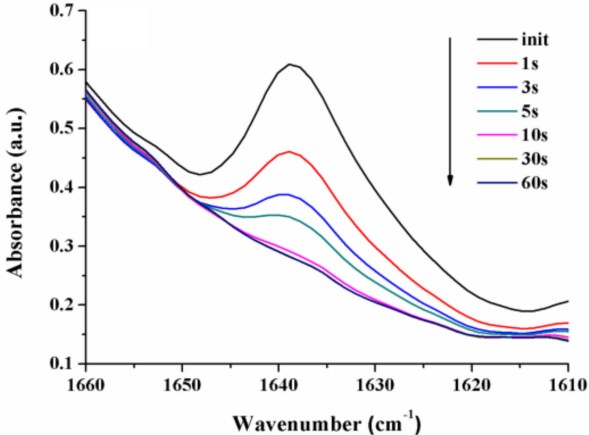

**Figure 5.** Modification of the double bond FTIR absorption band for L1 formulation during the irradiation with UV light.

The results indicated that the kinetics of the photopolymerization processes was scarcely affected by the incorporation of the inorganic particles. However, the low difference obtained for the conversion degree could be explained in terms of the restriction effect manifested by ZnO or ZnO-SnO$_2$ particles (Ps) on the mobility of the reactive centers, which led to the decrease of the methacrylic double-bond conversion degree [60]. In addition, the lower conversion values for both samples incorporating Ps could be due to the high ability of ZnO to capture the UV light [61,62].

Figure 6 illustrates the UV-Vis transmittance spectra of L1, L2, and L3 thin films. Accordingly, all films exhibited good transparency in the region stretching on the visible range. While the transmittance of L1 film was almost 85% for wavelengths beyond 350 nm, the same parameter for the films with 1wt.% ZnO and 1 wt.% ZnO-SnO$_2$ was beyond 80% and 70%, respectively. The slightly lower value of UV-Vis transparency for L3 film could be explained by the electron transitions between the valence and conduction bands. Also, the lower transparency of the film containing doped particles could be due to the UV and visible photons captured by the particles incorporated into the film [63].

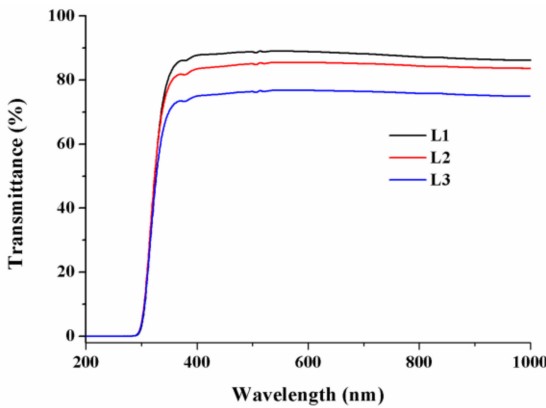

**Figure 6.** Transmittance spectra of L1–L3 thin films.

To thoroughly examine the morphology of the obtained polymeric films, they were broken in liquid nitrogen and then analyzed in the fracture section. The film homogeneity and particle distribution in the hybrid composites were assessed by scanning electron microscopy, as shown in Figure 7. The SEM image taken in fracture for the L1 sample, which had no particles in the composition (Figure 7a), indicated a homogenous structure without pores, the noticed irregularities being attributed to the way the film was broken. On the other hand, as observed in Figure 7b, the L2 formulation contained ZnO in the form of stars, homogeneously distributed in the polymer matrix, while the L3 fracture (Figure 7c) evidenced cube-like forms of ZnO doped with $SnO_2$ incorporated into the polymer matrix.

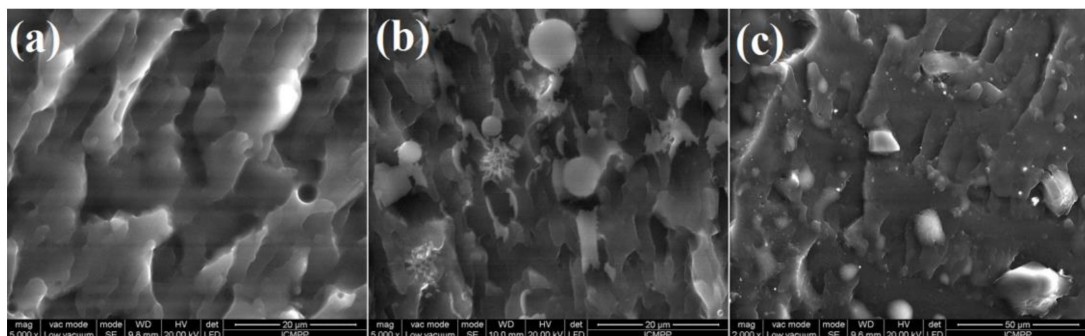

**Figure 7.** SEM images of polymeric films (in fracture) for L1 without particles (**a**), L2 with ZnO (**b**) and L3 with ZnO-$SnO_2$ (**c**).

The composition and spatial atoms distribution into the composite films based on L1, L2, and L3 formulations were surveyed by EDX pectroscopy and elemental dot mapping of the fractured films, which allowed us to asses the main atoms scattered in the samples (Figure 8).

Thus, the EDX spectrum of the L1 formulation was characterized by the presence of three peaks that corresponded to C, O, and N atoms (Figure 8a), in good agreement with the composition of the photopolymerized film without nanoparticles. For the L2 sample, besides the peaks for C, N, and O, some peaks characteristic for Zn atoms were clearly evidenced (Figure 8b), demonstrating that ZnO particles were successfully incorporated into the polymeric film. Moreover, the bright spots in the mapping image of Zn atoms indicated a uniform distribution of the atoms in the film. The EDX analysis of the L3 fractured film, illustrated in Figure 8c, denoted the presence of clear peaks characteristic for C, O, N, Zn, and Sn, a clear proof for the incorporation of ZnO-$SnO_2$ particles into the film during photopolymerization. Also, the uniform distribution of zinc and tin atoms inside the L3 photocrosslinked film could be remarked. The EDX analysis allowed the estimation of the content of Zn atoms to be as 0.88 wt.% in the L2 film, while in the L3 film, the content of the Zn and Sn atoms was found to be 0.96 wt.% and 0.92 wt.%, respectively.

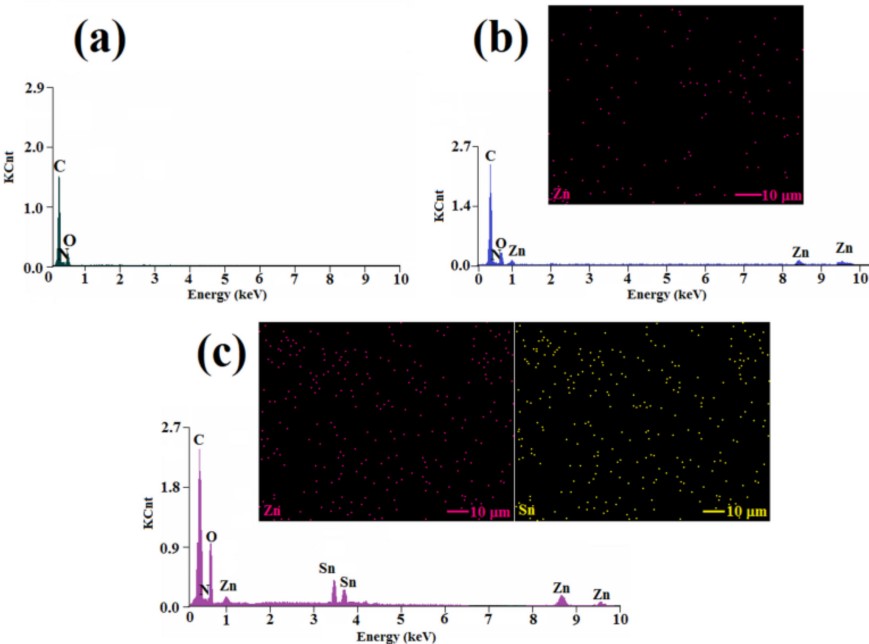

**Figure 8.** EDX analysis results for L1 (**a**) and the mapping images registered for L2 (**b**) and L3 (**c**) formulations.

## 2.4. Photocatalytic Activity

As mentioned in the literature, ZnO, in combination with other metal particles, can degrade various organic dyes in wastewater, including methylene blue (MB) [38] or rhodamine B [39], among others [64]. Consequently, we considered challenging to investigate the photocatalytic degradation of malachite green (MG) dye under irradiation with simulated solar light, by using L2 and L3 composite films as photocatalysts. To highlight the efficiency of photocatalysts L2 and L3, we firstly studied the photo-optical behavior of MG dye during the irradiation in the absence of any photocatalyst and when L1 film was used (Figure S3 SI). Although malachite green (MG) has a high photostability, it was observed that the absorption band at 616 nm was reduced with only 9% after 180 min of simulated solar light irradiation in the absence of any photocatalyst (Figure S3a). In the presence of L1 film (Figure S3b), there was a slight influence of about 3% (12% degradation after 60 min of irradiation) due to an absorption process, which produced a color change in this film from translucent white to a bluish color. This was more likely due to the absorption of the dye on the surface of the film, rather than a degradation process.

Figure 9 illustrates the evolution of the UV-Vis absorption spectra of MG dye in aqueous solutions at a concentration of $10^{-5}$ M, as was induced by composite films L2 and L3 under simulated solar light exposure, recorded after different irradiation times.

The characteristic absorption band of the MG dye at 616 nm registered an absorbance decrease of 43% after 60 min of irradiation when assisted by L2 film (Figure 9a), while it totally disappeared when L3 hybrid film was used as photocatalyst after only 45 min exposure to simulated solar light (Figure 9b), proving 100% photodegradation activity of L3 film with no signature for intermediary products (Figure S4, SI). This change could be viewed with the naked eye (Figure 9b, inset). These experimental features highlighted that the L3 film, containing doped $ZnO$-$SnO_2$ particles, exhibited higher photocatalytic performance than the L2 film, which contained only ZnO particles. The superior photocatalytic response of $ZnO$-$SnO_2$ could be associated with the cubic shape of the particles incorporated in the polymer matrix, increased light absorption, and reduced electron-hole pair recombination.

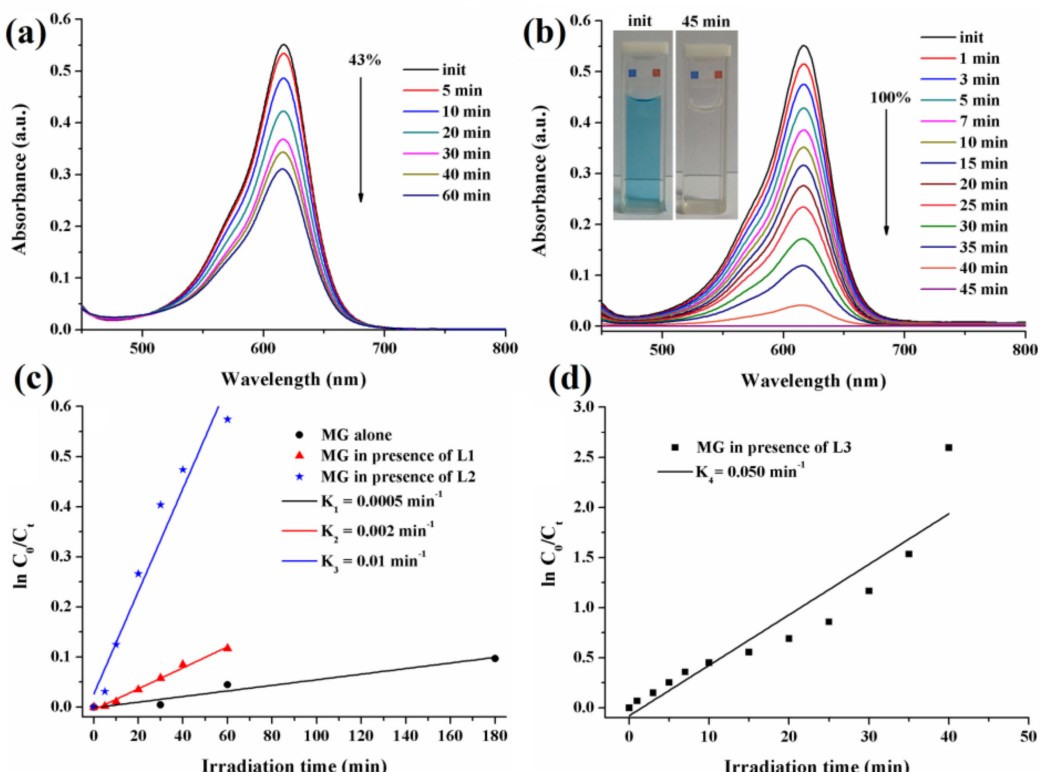

**Figure 9.** Irradiation-induced evolution of UV-Vis absorption spectra of malachite green (MG) solutions when L2 (**a**) and L3 (**b**) were used as photocatalysts and the corresponding kinetic curves for MG dye degradation (**c**,**d**).

To further asses the performance of L2 and L3 photopolymerized hybrid films as photocatalysts, the kinetic rate constant K was determined by using the formula $Kt = \ln C_0/C_t$, where t is the irradiation time (min), $C_0$ is the initial concentration of MG dye, and $C_t$ represents the concentration after t min of irradiation, by assuming that the process follows pseudo-first-order kinetics [30]. Figure 9c,d show the $\ln C_0/C_t$ plots as a function of the irradiation time. The kinetic rate constant for MG alone was $K_1 = 0.0005$ min$^{-1}$, and it increased to $K_2 = 0.002$ min$^{-1}$ for L1 film, $K_3 = 0.01$ min$^{-1}$ for L2 film, and $K_4 = 0.050$ min$^{-1}$ for L3 film. These results brought additional proof that the ZnO-SnO$_2$ cubes embedded in the polymer matrix exhibited a superior photocatalytic response compared to the equivalent samples containing ZnO star shapes.

The mechanism followed by different dyes during the photocatalytic decomposition is described in detail in the literature [65,66], as well as the reactions that occurred on the surface of the film [67].

To evaluate the reusability of the L3 composite film, repeated photocatalytic tests were performed using this sample. As illustrated in Figure 10, the photodegradation activity of the ZnO-SnO$_2$ composite was retained even after five consecutive cycles of reuse with an insignificant reduction of the photocatalytic performance. Such behavior suggested an excellent photostability of the photocatalyst. Photocrosslinked films were used since they maintain their integrity, and the particle leaching does not occur. Also, materials could be reused more easily than by using only simple particles, as previously demonstrated in the literature [38,39,68].

Additionally, to demonstrate that dye degradation took place as previously described, the $^1$H NMR spectra and total organic carbon (TOC) analysis of MG water solution before and after irradiation were registered. $^1$H NMR spectroscopy was used to identify the products that might appear during the degradation of MG. In Figure S5a, the appearance of the methyl group protons of MG at 1.28 ppm and 3.24 ppm were observed, while the signals corresponding to the aromatic protons were found between 5.82 ppm and 7.74 ppm. After visible irradiation in the presence of an L3 composite

film containing ZnO-SnO$_2$, the disappearance of the peaks corresponding to the MG dye along with solution discoloration occurred (Figure S5b), demonstrating that the degradation process took place. This also demonstrated that the film was not subjected to photodegradation and was stable under irradiation with simulated solar light.

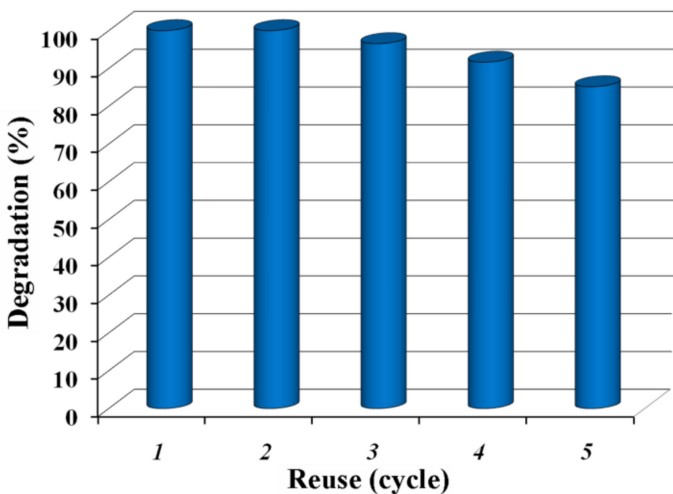

**Figure 10.** Reusability test for the L3 composite film (the estimated degradationin % corresponds to 45 min irradiation time).

The mineralization of MG dye and its decomposition to CO$_2$ was investigated by measuring the concentration of total organic carbon (TOC) in the solution irradiated with simulated solar light in the presence of the L3 film. The TOC analysis indicated mineralization of about 81%, the result suggesting that the present photocatalyst showed effective activity in the discoloration process and removal of dye organic carbon under simulated solar radiation.

## 3. Materials and Methods

### 3.1. Materials

Poly(propylene glycol) (M$_w$ =1000 g/mol), 2-isocyanatoethyl methacrylate, dibutyltin dilaurate, N,N-diisopropylcarbamoyl chloride, 2-hydroxyethyl methacrylate (HEMA), triethylamine, zinc nitrate hexahydrate, phenyl-bis(2,4,6-trimethylbenzoyl) phosphine oxide (Irg819), ammonia solution (25%), tin(IV) chloride pentahydrate, malachite green (MG), tetrahydrofuran (THF), methylene chloride, and deionized water were provided from Sigma-Aldrich, (Merck, Germany) and used without further purifications.

### 3.2. Synthesis

#### 3.2.1. Poly(propylene glycol) Dimethacrylate (PO-UDMA)

Poly(propylene glycol) dimethacrylate (PO-UDMA) was prepared following a published procedure [38]. Briefly, in a three-necked round bottom flask equipped with a magnetic stirrer, poly(propylene glycol) (4 mmol) and 2-isocyanatoethyl methacrylate (8 mmol) were mixed in 10 mL anhydrous THF, by using dibutyltin dilaurate as a catalyst. The product evolution was monitored by FTIR, following the disappearance of the absorption band characteristic for the isocyanate unit at ~2260 cm$^{-1}$. The reaction was continuously stirred for 24 h at 38 °C, and after that, the solvent was removed under reduced pressure. Finally, the monomer was obtained as a viscous liquid.

### 3.2.2. N,N-(diisopropylcarbamoyloxy)Ethyl Methacrylate (N-MA)

In a 50 mL three-necked round-bottomed flask equipped with a magnetic stirrer, a dropping funnel, and a thermometer, 5 g (31.17 mmol) of N,N-diisopropylcarbamoyl chloride and 4.33 mL (31.17 mmol) of triethylamine were placed in 10 mL methylene chloride. Then, the reaction was allowed to cool to 0 °C, and 3.82 mL (31.17 mmol) of HEMA in 5 mL methylene chloride was dropped with stirring for 1 h. After 24 h of the mixture at room temperature, the liquid monomer was obtained after solvent removal under reduced pressure.

FTIR (KBr, cm$^{-1}$): 2971-2877 (aliphatic C–H stretching); 1723 (C=O stretching), 1638 and 815 (C=C stretching); 1291 (C-O-C stretching).

$^{13}$C NMR in CDCl$_3$ ($\delta$, ppm): 166.16 and 154.38 (>C=O), 135.08 and 124.88 (>C=CH$_2$), 61.96 and 59.68 (-CH$_2$-), 46.41-44.87 (-CH-), 19.57 and 17.35 (CH$_3$).

### 3.2.3. ZnO Particles

This synthesis was carried out in a 500 mL three-necked flask, by using 13.9 mmol of zinc nitrate hexahydrate and 350 mL deionized water. The pH was allowed to reach 10-11 by using ammonia (25%) when the solution evolved from turbid to clear. Then, the resulting mixture was refluxed under vigorous stirring for 3 h to allow the evolution of ZnO particles. The solid product was separated by centrifugation, followed by washing with deionized water and drying for 1 day at room temperature. Then, the as-resulted solid was calcinated in the air at 500 °C for 3 h, when ZnO particles were achieved in the form of white powder (bandgap for ZnO particles is 3.3 eV, values were calculated according to the literature [69,70]).

### 3.2.4. ZnO-Doped SnO$_2$ Particles

Doped ZnO particles were prepared using the identical procedure described above. A molar ratio of 1:1 zinc nitrate hexahydrate (Zn) to tin (IV) chloride pentahydrate (Sn) was used. After isolation by centrifugation and washing with deionized water, the product was dried for 1 day at room temperature and calcinated at 500 °C for 3 h. This sample was abbreviated as ZnO-SnO$_2$ (bandgap for ZnO-SnO$_2$ particles is 2.9 eV, values were calculated according to the literature [69,70]).

### 3.3. Films Preparation

The polymer films used in this study were synthesized through the photopolymerization of monomer mixtures, using Irg819 as photoinitiator, the experimental composition for films being presented in Table 1. In order to achieve homogeneity, several drops of acetone were introduced, and then the mixtures were cast on glass or Teflon plates and photopolymerized. The irradiation was carried out with a Hg–Xe lamp (Hamamatsu Lightningcure Type LC8, Model L9588) at 365 nm, having a light intensity of 40 mW/cm$^2$.

### 3.4. Photocatalytic Activity Investigation

The photopolymerized films were studied with regard to their photocatalytic efficiency by investigating the degradation process of malachite green (MG) in aqueous solutions (pH = 5), under continuous stirring, in ambient conditions and under irradiation with light from visible range. The same device, Hamamatsu Lightningcure Type LC8, Model L9588 was used for the degradation experiments, only that the lamp was changed to a Xe type visible light source ($\lambda$ = 400–800 nm, maximum at 437 nm, UV contribution of ~ 5 mW/cm$^2$), which mimics sunlight. Apart from that, each photopolymerized film (~1 g) was introduced into 50 mL of MG solutions. This mixture was irradiated by a programmed time regime with the visible light source coming from the Xe lamp. At a given time interval, 1.5 mL of the solution was taken and further analyzed by UV-vis spectrometry. The experimental set-up was added to the Supporting Information (SI) section, Figure S6.

*3.5. Measurements*

The structure of the synthesized monomers was identified by FTIR, [1]H, and [13]C NMR spectroscopies. [1]H and [13]C NMR spectra were registered on a Bruker 400MHz spectrometer at room temperature (BRUKER, Karlsruhe, Germany). The NMR experiments were recorded with a 5 mm four nuclei direct detection z-gradient probe. Chemical shifts were reported in δ units (ppm) and were referenced to the residual solvent signal ($D_2O$ at 4.78 ppm for [1]H). NMR spectra were recorded using standard pulse sequence as delivered by Bruker with TopSpin 4.0.5 spectrometer control and processing software. All the spectra were registered at room temperature (24 °C) in $D_2O$ water at pH = 5, with acquisition parameters ns = 64. The Fourier transform infrared (FTIR) spectra were recorded on a Bruker Vertex 70 FTIR spectrometer (BRUKER, Karlsruhe, Germany). To evaluate the conversion degree by FTIR analysis, homogeneous mixtures of photopolymerizable monomers were synthesized by following the data listed in Table 1 and then manually coated on KBr pellets and irradiated with UV light. The FTIR absorption spectra were registered at different irradiation times, and the modification, which appeared in the C=C stretching vibration at 1638 $cm^{-1}$, was used to estimate the degree of conversion. The irradiation of the monomer mixtures containing ZnO or ZnO-$SnO_2$ was carried out by using the UV light with an intensity of 40 mW/$cm^2$ generated by a Hg-Xe lamp (Hamamatsu Lightningcure Type LC58, Model L9588, HAMAMATSU PHOTONICS K.K., Iwata City, Shizuoka Pref., Japan). The UV-Vis absorption spectra were obtained on a Perkin Elmer Lambda 2 spectrophotometer (PerkinElmer Inc., Wellesley, MA, United States). XRD spectra were measured on a Bruker D8 ADVANCE (BRUKER, Karlsruhe, Germany) diffractometer by using Cu Kα radiation (λ = 0.1541 nm). The spectral patterns were registered from 20° to 100°, at a scan rate of 0.02° $min^{-1}$. Raman spectroscopy experiments were performed by using a Renishaw in Via confocal microscope equipped with a He-Ne laser at 632.8 nm (17 mW) and a CCD detector coupled to a Leica DM 2500 M microscope. All experiments were performed in backscattering geometry, using a 100× objective with a numerical aperture of 0.75, at room temperature and atmospheric pressure. All spectra were read with the WiRE 3.2 software (Renishaw, New Mills, GL, UK). TEM and SEM micrographs were obtained at 120 kV, by using a Hitachi High-Tech HT7700 (Hitachi High-Tech GLOBAL, Tokyo, Japan) instrument and an environmental scanning electron microscope QUANTA200 coupled with an energy-dispersive X-ray spectroscope (ESEM/EDX), respectively (FEI Company, Netherlands). The total organic carbon (TOC) of the MG solution before and after visible irradiation in the presence of L3 film was measured by using a Multi N/C 2100 Analyticjena analyzer (Analytik Jena AG, Jena, Germany).

## 4. Conclusions

A new monomer, namely, N,N-(diisopropylcarbamoyloxy)ethyl methacrylate (N-MA), was synthesized and used together with PO-UDMA and metal nanoparticles in the development of hybrid composites by photopolymerization. To this scope, flower-shaped ZnO and unique cube-shaped ZnO-$SnO_2$ particles were prepared and subsequently incorporated into photopolymerizable formulations. According to FTIR spectra, the monomers displayed a good photoreactivity during the photopolymerization process, regardless of its occurrence in the absence or in the presence of metal particles. The particles were investigated with respect to their size, shape, and morphology by using several methods, such as ESEM/EDX, TEM, XRD, and Raman analyses, all proving their shape and purity. The incorporation and homogeneous distribution of ZnO and ZnO-$SnO_2$ particles in the organic phase were fully confirmed by ESEM/EDX measurements. The photocatalytic activity tests highlighted that the prepared cube-shaped ZnO-$SnO_2$-based photocatalyst displayed an excellent photocatalytic activity. After only 45 min of irradiation, the discoloration efficiency of MG dye in aqueous solutions achieved 100% with this photocatalyst, while for the flower-shaped ZnO composite film, degradation of 43% was obtained after 60 min of irradiation. Overall, the above results indicated that the newly prepared film based on cubic ZnO-$SnO_2$ was an efficient photocatalyst and could be considered for future treatment of wastewater containing dyes.

**Supplementary Materials:** The following are available online at http://www.mdpi.com/2076-3417/10/6/1954/s1.

**Author Contributions:** Writing—original draft preparation, V.-E.P., writing—review and editing, V.-E.P. and M.-D.D., supervision, M.-D.D. All authors have read and agreed to the published version of the manuscript.

**Funding:** This work was supported by a grant of the Ministry of Research and Innovation, CNCS - UEFISCDI, project number PN-III-P1-1.1-PD-2016-1718 (36/2018), within PNCDI III.

**Conflicts of Interest:** The authors declare no conflict of interest.

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
