# Peer review of "Photopolymerized Films with ZnO and Doped ZnO Particles Used as Efficient Photocatalysts in Malachite Green Dye Decomposition"

_applsci, doi:10.3390/app10061954_

Round 1
Reviewer 1 Report
Major revision.
Throughout all the work the authors speak about “doped ZnO”, but there are no characterization evidence that indicates that Sn is introduced into the structure of ZnO (either in interstitial or substitutional positions). Rather, there is a composite with two independent phases of Zn oxide and Sn oxide. In fact, the authors themselves name the alleged "doped" sample as ZnO-SnO2. In fact, in the diffraction patterns shown in Figure 2, peaks assignable to an SnO2 phase appear, thus pointing to the fact that there is no doping with Sn. In any case, from those XRD patterns, one can hardly say that crystalline phases are present. Only a few small peaks can be assignable to an SnO2 phase and nothing can be said about the Zn phase, indicating rather the presence of amorphous phases. It could be that there was no crystalline domain with enough long-distance order to give diffraction. However, the Raman spectrum much more sensitive to the structure also indicates that there are no crystalline phases.
The authors cannot say that they have wurzite ZnO phase and SnO2 phase with that Raman spectrum (red color figure 2b). In fact, they rely exclusively on the observation of a small shoulder at 580 cm-1. It is known that Raman spectrum of ZnO hexagonal structure is characterized by the active modes A1, E1 and E2. The most characteristic peak of this phase appears at 433-453 cm-1 and can be assigned to E2H mode. Other peaks related to the E2 mode are the 90 cm-1 assignable to the E2L, the 195cm-1 to the 2E2L and the 331-327 cm-1 to the E2H-E2L. Mode A1 only shows its peak at 374 cm-1 corresponding to the transverse (A1T). However, the peak at 580 cm-1 corresponding to the longitudinal (E1L) is associated with the presence of structural defects. Therefore, in the authors data, if the characteristic peak at 430-450 cm -1 is not seen, it cannot be said that the 580 peak indicates that there is a hexagonal phase of ZnO.
Similarly in Figure S2 in the FTIR spectrum no bands assignable to ZnO are seen.
Therefore, first it is not a doped sample and second, it doesn’t present crystalline phases. Rather it is amorphous.
Other issues:
- Explain more in the introduction on the polymerization method to create films and the impact of using films.
- The introduction should explain better why the choice of tin, which is the strategy of combining it with the ZnO. The cited references of Sn are referred to the use of the Sn oxide alone, not combinations. Also they should define their aim: create a doped structure or create a composite. In the literature there are many examples of both cases.
- Change the title 2.1. Specify that it is the characterization of the monomer (no all the materials)
- The numbering of the Figures is wrong: between Figure 2 and 3 there is one labelled S2
- Figure S2b shows the FTIR spectra of the films but there is not mention or description in the text. Explain the result and discuss it.
- Line 147: it is not the holder, it is the grid.
- The EDX maps should be accompanied with the image from where they were done. Specify if those are TEM or EDX images (from the magnification it seems probably SEM specify).
- On line 212-213 it is said “the mapping image of Zn atoms indicated through the presence of bright spots a uniform distribution of the atoms in the 213 film.” It is not a mapping of atoms, each spot does not correspond to an atom. It means that in that area the presence of that element is detected, but in reality that area depends on the magnification and size of the spot; assuming that the EDX correspond to the SEM images shown in Figure 7, each spot corresponds to an area of ​​micron size or a lot of nanometers. On another hand, with that resolution it is not possible to say whether they have something doped or a kind of separated phases again
- Line 218-219: how have they calculated these contents from the EDX? It should be described in the experimental part.
- The values ​​obtained are incompatible with those expressed in table 1. If the Zn:Sn ratio is 1:1, how do they get the same weight content?(line 219). If there is 1% wt of ZnO-SnO (table 1), they cannot find almost 1%wt of Zn and for Sn.
- They should calculate the band-gap of the materials.
- Line 252: correct the error. It is not Kt = ln C0/Ct, it is: K.t = - C0/Ct
- Figure 10. The conversion decrease is appreciable ~ 15%. Therefore, it is not possible to say “insignificant reduction of the photocatalytic performance”. How do they justify this variation if there is no leaching of the active components?
- Line 346: how do they select the wavelength, with bandpass filters?
- Line 351. Indeed, a Xe-lamp is a solar light simulator. The doubt is: did they use a bandpass filter to select the 437 nm wavelength? If they did not use a filter and the light emitted by the lamp is irradiated directly, the light is not pure visible, it has a contribution of ultraviolet light (it is only 5% but that contribution is important). If this is the case, they should change along all the text “visible light” by sunlight . If, on the contrary, the light was purely visible, the argument on line 262-263 is wrong, since there is not enough energy to excite in ZnO, its bandgap is 3.2 eV, therefore there are no electrons in the conduction band of the ZnO.
Author Response
Dr. Viorica-Elena Podasca
Romanian Academy
“Petru Poni” Institute of Macromolecular Chemistry
41 A Gr. Ghica Voda Alley, 700487 Iasi-Romania
February 28, 2020
Answers to Reviewer 1:
Dear Professor,
Thank you for your effort in reviewing our paper entitled “Photopolymerized Films with ZnO and Doped ZnO Particles Used as Efficient Photocatalysts in Malachite Green Dye Decomposition”, authors Viorica-Elena Podasca and Mariana-Dana Damaceanu (manuscript ID: applsci-727140). After a deep examination of the manuscript and tacking into consideration the reviewer’s comments, it was accordingly revised, as follows (in red color).
Comment 1: Throughout all the work the authors speak about “doped ZnO”, but there are no characterization evidence that indicates that Sn is introduced into the structure of ZnO (either in interstitial or substitutional positions). Rather, there is a composite with two independent phases of Zn oxide and Sn oxide. In fact, the authors themselves name the alleged "doped" sample as ZnO-SnO2. In fact, in the diffraction patterns shown in Figure 2, peaks assignable to an SnO2 phase appear, thus pointing to the fact that there is no doping with Sn. In any case, from those XRD patterns, one can hardly say that crystalline phases are present. Only a few small peaks can be assignable to an SnO2 phase and nothing can be said about the Zn phase, indicating rather the presence of amorphous phases. It could be that there was no crystalline domain with enough long-distance order to give diffraction. However, the Raman spectrum much more sensitive to the structure also indicates that there are no crystalline phases.
The authors cannot say that they have wurzite ZnO phase and SnO2 phase with that Raman spectrum (red color figure 2b). In fact, they rely exclusively on the observation of a small shoulder at 580 cm-1. It is known that Raman spectrum of ZnO hexagonal structure is characterized by the active modes A1, E1 and E2. The most characteristic peak of this phase appears at 433-453 cm-1 and can be assigned to E2H mode. Other peaks related to the E2 mode are the 90 cm-1 assignable to the E2L, the 195cm-1 to the 2E2L and the 331-327 cm-1 to the E2H-E2L. Mode A1 only shows its peak at 374 cm-1 corresponding to the transverse (A1T). However, the peak at 580 cm-1 corresponding to the longitudinal (E1L) is associated with the presence of structural defects. Therefore, in the authors data, if the characteristic peak at 430-450 cm -1 is not seen, it cannot be said that the 580 peak indicates that there is a hexagonal phase of ZnO.
Similarly in Figure S2 in the FTIR spectrum no bands assignable to ZnO are seen.
Therefore, first it is not a doped sample and second, it doesn’t present crystalline phases. Rather it is amorphous.
Reply 1: We agree with R1 opinion that the ZnO-SnO2 is an amorphous state, as was mentioned in the manuscript at Particle characterization section, lines 108-109. The assignments of the peaks in XRD, FTIR and Raman spectra were made by comparing with literature data.
- The Raman band at 560 cm-1 is broader than that of the pure ZnO sample; therefore we concluded that the bands characteristic for Zn-O and Sn-O overlap. Similar spectra were found for Sn-doped ZnO, as reported in references 47 [Ozgur M.; Pat S.; Mohammadigharehbagh R.; Musaoglu C.; Demirkol U.; Elmas S.; Ozen S.; Korkmaz S. Sn doped ZnO thin film deposition using thermionic vacuum arc technique. J. Alloy. Comp. 2019, 774, 1017-1023.] and 48 [Ganesh V.; Yahia I.S.; AlFaify S.; Shkir M. Sn-doped ZnO nanocrystalline thin films with enhanced linear and nonlinear optical properties for optoelectronic applications. J. Phys. Chem. Solids 2017, 100, 115–125.].
Comment 2: Explain more in the introduction on the polymerization method to create films and the impact of using films.
- The introduction should explain better why the choice of tin, which is the strategy of combining it with the ZnO. The cited references of Sn are referred to the use of the Sn oxide alone, not combinations. Also they should define their aim: create a doped structure or create a composite. In the literature there are many examples of both cases.
Reply 2: - Our objective was to synthesize doped ZnO particles and to incorporate them into a polymer matrix for better reusability. The literature data report many studies focused on powders which require tedious work in the recycling processes. In our case, photocrosslinked films were used since they maintain the integrity and the particles leaching does not occur. In addition, these materials can be reused more easily than the simple particles, as previously demonstrated by other studies. We already mentioned these aspects in the manuscript, at page 9, rows 272-274.
- The reason for choosing the SnO as doping partner is mentioned in the Introduction section, at page 2, rows 67-71.
Comment 3: - Change the title 2.1. Specify that it is the characterization of the monomer (no all the materials)
- The numbering of the Figures is wrong: between Figure 2 and 3 there is one labelled S2
Reply 3: - The title of section 2.1 was changed.
- Thank you for this note. Due to some changes made by the editorial, some issues related to figure position and numbering occurred. In the revised manuscript, the Supporting Information was placed at the end of the manuscript, after the references list; thus all the figures are now as previously planned.
Comment 4: - Figure S2b shows the FTIR spectra of the films but there is not mention or description in the text. Explain the result and discuss it.
- Line 147: it is not the holder, it is the grid.
Reply 4: - Since the FTIR spectra did not provided relevant data we completely removed Figure S2b from the manuscript and SI.
- For SEM analysis, a carbon band was used to fix the sample (which is a powder), therefore in discussion is a holder. The grid was used in TEM analysis.
Comment 5: - The EDX maps should be accompanied with the image from where they were done. Specify if those are TEM or EDX images (from the magnification it seems probably SEM specify).
- On line 212-213 it is said “the mapping image of Zn atoms indicated through the presence of bright spots a uniform distribution of the atoms in the 213 film.” It is not a mapping of atoms, each spot does not correspond to an atom. It means that in that area the presence of that element is detected, but in reality that area depends on the magnification and size of the spot; assuming that the EDX correspond to the SEM images shown in Figure 7, each spot corresponds to an area of micron size or a lot of nanometers. On another hand, with that resolution it is not possible to say whether they have something doped or a kind of separated phases again
- Line 218-219: how have they calculated these contents from the EDX? It should be described in the experimental part.
- The values obtained are incompatible with those expressed in table 1. If the Zn:Sn ratio is 1:1, how do they get the same weight content?(line 219). If there is 1% wt of ZnO-SnO (table 1), they cannot find almost 1%wt of Zn and for Sn.
Reply 5: - The EDX map was acquired from the SEM analysis. This was explained in section 3.5. Measurements: “environmental scanning electron microscope QUANTA200 coupled with an energy-dispersive X-ray spectroscope (ESEM/EDX)”
- The results are provided from in a certain area of the material. EDX can be used for qualitative (the type of elements) as well as quantitative (the percentage of the concentration of each element of the sample) analysis. In most SEMs, dedicated software enables auto-identification of the peaks and calculation of the atomic percentage of each element that is detected. Usually the software computes these concentrations by comparing the intensities of X-ray peaks of different elements. As already mentioned, soft X-rays (energy below 3 keV) can be absorbed and therefore, quantification of elements using soft X-ray lines could lead to some errors when the sample is thicker than few hundreds of nm. Therefore we stated that “an estimation” was given by EDX analysis.
Comment 6: - Line 252: correct the error. It is not Kt = ln C0/Ct, it is: K.t = - C0/Ct
- Figure 10. The conversion decrease is appreciable ~ 15%. Therefore, it is not possible to say “insignificant reduction of the photocatalytic performance”. How do they justify this variation if there is no leaching of the active components?
- Line 346: how do they select the wavelength, with bandpass filters?
- Line 351. Indeed, a Xe-lamp is a solar light simulator. The doubt is: did they use a bandpass filter to select the 437 nm wavelength? If they did not use a filter and the light emitted by the lamp is irradiated directly, the light is not pure visible, it has a contribution of ultraviolet light (it is only 5% but that contribution is important). If this is the case, they should change along all the text “visible light” by sunlight . If, on the contrary, the light was purely visible, the argument on line 262-263 is wrong, since there is not enough energy to excite in ZnO, its bandgap is 3.2 eV, therefore there are no electrons in the conduction band of the ZnO.
Reply 6: - For our study we used the formula Kt= ln Co/Ct according to reference 30 [Jin C.; Ge C.; Jian Z.; Wei Y. Facile Synthesis and High Photocatalytic Degradation Performance of ZnO-SnO2 Hollow Spheres. Nanoscale Res. Lett. 2016, 11:526.].
- The film was tested in 5 consecutive cycles in the same day to test its robustness and degradation activity towards MG. The fact that the film was not fully regenerated after each test can explain the almost 15 % loss of the catalytic activity after 5 cycles.
- The wavelength of the lamp is predetermined visible light source (Xe lamp, λ = 400–800 nm). The maximum of the lamp is 437 nm, but it also has an UV contribution of ~ 5 mW/cm2, just like sunlight. For more details please see the Hamamatsu Lightning cure Type LC58, Model L9588, Xe lamp device characteristics.
- To use the term “sunlight” seems improper to us since we didn’t made the experiments under the real sun. On our knowledge, the light emitted by our lamp is accepted as “visible light”, therefore we consider this term more appropriate.

Reviewer 2 Report
Podasca and Damaceanu reported on the composites material (inorganic and organic material), which is a good way to overcome some limitations in the single materials. The work is fine and can be published at the Applied Sciences journal after language check and presenting the supplementary information. It is not available in the link. Please also mention clearly the main differences between the traditional materials in these new composites materials.
Author Response
Dr. Viorica-Elena Podasca
Romanian Academy
“Petru Poni” Institute of Macromolecular Chemistry
41 A Gr. Ghica Voda Alley, 700487 Iasi-Romania
February 28, 2020
Answers to Reviewer 2:
Dear Professor,
Thank you for your effort in reviewing our paper entitled “Photopolymerized Films with ZnO and Doped ZnO Particles Used as Efficient Photocatalysts in Malachite Green Dye Decomposition”, authors Viorica-Elena Podasca and Mariana-Dana Damaceanu (manuscript ID: applsci-727140). After a deep examination of the manuscript and tacking into consideration the reviewer’s comments, it was accordingly revised, as follows (in red color).
Comment: Podasca and Damaceanu reported on the composites material (inorganic and organic material), which is a good way to overcome some limitations in the single materials. The work is fine and can be published at the Applied Sciences journal after language check and presenting the supplementary information. It is not available in the link. Please also mention clearly the main differences between the traditional materials in these new composites materials.
Reply: - Due to some changes made by the editorial, the SI was mixed with text from the manuscript. In the revised manuscript, the Supporting Information was placed at the end of the manuscript.
- The difference between the traditional materials and our composites are highlighted in the Introduction section. The literature data reports many studies focused on powders which require tedious work in the recycling processes. In our case, photocrosslinked films were used since they maintain the integrity and the particles leaching does not occur. In addition, these materials can be reused more easily than the simple particles, as previously demonstrated by various studies. We already mentioned these aspects in the manuscript, at page 9, rows 271-274.
- The language of the manuscript was revised.

Reviewer 3 Report
In this manuscript is described the synthesis of ZnO and ZnO doped with tin oxide (ZnO-SnO2) in form of particles and successively incorporated into a polymeric matrix by photopolymerization reaction of N,N-(diisopropylcarbamoyloxy)ethyl methacrylate (N-MA) together with poly(propylene glycol)dimethacrylate (PO-UDMA) for the production of ZnO and ZnO-SnO2 based hybrid composites by photopolymerization.
The photopolymerized films with ZnO and ZnO-SnO2 particles were successively tested in the photocatalytic degradation of Malachite Green dye (MG).
The materials used in this work have been well characterized by FTIR spectroscopy, X-ray diffraction, environmental scanning electron microscopy, transmission electron microscopy and Raman spectroscopy.
Specific comments are following reported:
- More detail about the process conditions used to perform the photocatalytic experiments should be given (pH value, stirring, temperature, controlled atmosphere -under air?- ).
- A new figure showing the experimental set-up could useful for the reader.
-The authors have stated that "photocrosslinked films were used since they maintain their integrity and the particles leaching do not occur".
How have they demonstrated the absence of ZnO or SnO2 leaching?
In the case of the most active hybrid film containing (ZnO-SnO2)...... Did the authors considered a possible ion formation is solution [e.g. Zn(II) or Sn(IV)] in relation with the experimental conditions? (to this purpose see also the comment related to the pH value)
- Please, specify the experimental conditions of the experiment carried out in D2O.
- Total Organic Carbon measurements are also suggested in order to confirm the effective degradation of MG excluding the release of organic species in solution deriving from the polymer matrix.
Publication of this manuscript is recommended after minor revisions.
Author Response
Dr. Viorica-Elena Podasca
Romanian Academy
“Petru Poni” Institute of Macromolecular Chemistry
41 A Gr. Ghica Voda Alley, 700487 Iasi-Romania
February 28, 2020
Answers to Reviewer 3:
Dear Professor,
Thank you for your effort in reviewing our paper entitled “Photopolymerized Films with ZnO and Doped ZnO Particles Used as Efficient Photocatalysts in Malachite Green Dye Decomposition”, authors Viorica-Elena Podasca and Mariana-Dana Damaceanu (manuscript ID: applsci-727140). After a deep examination of the manuscript and tacking into consideration the reviewer’s comments, it was accordingly revised, as follows (in red color).
Comment 1: In this manuscript is described the synthesis of ZnO and ZnO doped with tin oxide (ZnO-SnO2) in form of particles and successively incorporated into a polymeric matrix by photopolymerization reaction of N,N-(diisopropylcarbamoyloxy)ethyl methacrylate (N-MA) together with poly(propylene glycol)dimethacrylate (PO-UDMA) for the production of ZnO and ZnO-SnO2 based hybrid composites by photopolymerization. The photopolymerized films with ZnO and ZnO-SnO2 particles were successively tested in the photocatalytic degradation of Malachite Green dye (MG). The materials used in this work have been well characterized by FTIR spectroscopy, X-ray diffraction, environmental scanning electron microscopy, transmission electron microscopy and Raman spectroscopy.
Specific comments are following reported:
- More detail about the process conditions used to perform the photocatalytic experiments should be given (pH value, stirring, temperature, controlled atmosphere -under air?- ).
Reply 1: - The photocatalytic experiments were conducted in aqueous solutions (pH = 5), in ambient conditions, under visible light irradiation and continuous stirring. These conditions are detailed in section 3, subsection 3.4.
Comment 2: - A new figure showing the experimental set-up could useful for the reader.
Reply 2: - A new figure with the experimental set-up was added to the Supporting Information (SI) section.
Comment 3: - The authors have stated that "photocrosslinked films were used since they maintain their integrity and the particles leaching do not occur".
How have they demonstrated the absence of ZnO or SnO2 leaching?
In the case of the most active hybrid film containing (ZnO-SnO2)...... Did the authors considered a possible ion formation is solution [e.g. Zn(II) or Sn(IV)] in relation with the experimental conditions? (to this purpose see also the comment related to the pH value)
Reply 3: - The statement "photocrosslinked films were used since they maintain their integrity and the particles leaching do not occur" was demonstrated and a comment was made in the manuscript (pages 9 and 10, lines 278-287) and SI (Figure S6)
- We assumed that in solution, the formation of specified ions is not probable, mainly based on the fact that the film can be reused multiple times without significant loss of the photocatalytic activity. At pH = 5, there is no probability for ion formation in water.
Comment 4: - Please, specify the experimental conditions of the experiment carried out in D2O.
- Total Organic Carbon measurements are also suggested in order to confirm the effective degradation of MG excluding the release of organic species in solution deriving from the polymer matrix.
Publication of this manuscript is recommended after minor revisions.
Reply 4: - The NMR experiments were recorded with a 5 mm four nuclei direct detection z-gradient probe. Chemical shifts are reported in δ units (ppm) and were referenced to the residual solvent signal (D2O at 4.78 ppm for 1H). NMR spectra were recorded using standard pulse sequence as delivered by Bruker with TopSpin 4.0.5 spectrometer control and processing software. All the spectra were recorded at room temperature (24 ºC), in D2O water at pH = 5, with acquisition parameters of ns = 64. Al these details are given in the Measurements section.
- The mineralization of MG dye and its decomposition to CO2 was investigated by measuring the concentration of total organic carbon (TOC) in the solution irradiated with visible light in the presence of the L3 film. The TOC analysis indicated a mineralization of about 81%, the result suggesting that L3 photocatalyst showed effective activity in bleaching process and in the removal of dye organic carbon under simulated solar radiation. This information is given in the manuscript at page 10, rows 288-292.

Reviewer 4 Report
this work looks very interesting and many results have been reported.
I believe it can be published after a minor revision
for authors the following suggestions:
It is necessary to improve the introduction and the comparison with literature. ZnO is a semiconductor used in photocatalysis for several purposes. I suggest to report some of the recent papers regarding the application of this photocatalyst, also coupled with other semiconductors, for example:10.3303/CET1652142
The absorbance spectra of the MG are interesting .. however they are reported only in the range 400-800 nm. The authors should report the full absorbance spectrum, even in the UV range. The UV range can give information on the presence of any intermediates, and can guarantee the complete degradation of the molecule.
Furthermore, the recyclability tests are very interesting. However, I would suggest evaluating the possible degradability of the polymer, reporting data that confirms not only the ability to reuse the catalyst but also the stability of the support.
Author Response
Dr. Viorica-Elena Podasca
Romanian Academy
“Petru Poni” Institute of Macromolecular Chemistry
41 A Gr. Ghica Voda Alley, 700487 Iasi-Romania
February 28, 2020
Answers to Reviewer 4:
Dear Professor,
Thank you for your effort in reviewing our paper entitled “Photopolymerized Films with ZnO and Doped ZnO Particles Used as Efficient Photocatalysts in Malachite Green Dye Decomposition”, authors Viorica-Elena Podasca and Mariana-Dana Damaceanu (manuscript ID: applsci-727140). After a deep examination of the manuscript and tacking into consideration the reviewer’s comments, it was accordingly revised, as follows (in red color).
Comment 1: this work looks very interesting and many results have been reported.
I believe it can be published after a minor revision
for authors the following suggestions:
It is necessary to improve the introduction and the comparison with literature. ZnO is a semiconductor used in photocatalysis for several purposes. I suggest to report some of the recent papers regarding the application of this photocatalyst, also coupled with other semiconductors, for example:10.3303/CET1652142
Reply 1: - Some changes were made to the Introduction section and the suggested reference was cited.
Comment 2: The absorbance spectra of the MG are interesting .. however they are reported only in the range 400-800 nm. The authors should report the full absorbance spectrum, even in the UV range. The UV range can give information on the presence of any intermediates, and can guarantee the complete degradation of the molecule.
Reply 2: - In SI, figure S5 we provided the absorption spectrum in the range of 220-800 nm. As can be observed, no intermediate compounds appeared, while the absorption bands completely decreased in the presence of L3 polymer film, being an indicative for no intermediates.
Comment 3: Furthermore, the recyclability tests are very interesting. However, I would suggest evaluating the possible degradability of the polymer, reporting data that confirms not only the ability to reuse the catalyst but also the stability of the support.
Reply 3: - We mentioned in the manuscript, pages 9-10, rows 278-287, based on NMR experiments, the film is not subjected to photodegradation and is stable under visible light irradiation.

Round 2
Reviewer 1 Report
Reply 1 from authors: We agree with R1 opinion that the ZnO-SnO2 is an amorphous state, as was mentioned in the manuscript at Particle characterization section, lines 108-109. The assignments of the peaks in XRD, FTIR and Raman spectra were made by comparing with literature data.
Reviewer comment: In lines 107-108 they said “Thus, due to the overlap of the peaks, it can be assumed that the ZnO-SnO2 sample is in an amorphous phase [50-52].” But it is no due to overlap of peak, is due to all the reason exposed in 1st revision. Overlapping peaks can make difficult the assignment, but it doesn’t justify amorphous. Rewrite that discussion
Reply 2 from authors: - The reason for choosing the SnO as doping partner is mentioned in the Introduction section, at page 2, rows 67-71.
Reviewer comment: In those lines they say Sn oxide is good in several applications. And that’s all. But if their aim was to “dope” (what they didn’t get) they should explain what they expect from doping: structural modifications, electronic changes, bandgap modification etc
Reply 6 from authors: To use the term “sunlight” seems improper to us since we didn’t made the experiments under the real sun. On our knowledge, the light emitted by our lamp is accepted as “visible light”, therefore we consider this term more appropriate.
Reviewer comment: Xe lamp is known commercially and widely in the literature as a solar light simulator. Therefore, they may not use the term sunlight and use "simulated solar ligh" instead
It is not necessary to go to the specifications of the brand of your lamp. It is known that the Xe spectrum has a UV contribution. In addition, the contribution of 5% UV is not negligible and can also intervene in the catalytic process.
And this is also important for the next point:
Comment in 1st revision: “Line 351. Indeed, a Xe-lamp is a solar light simulator. The doubt is: did they use a bandpass filter to select the 437 nm wavelength? If they did not use a filter and the light emitted by the lamp is irradiated directly, the light is not pure visible, it has a contribution of ultraviolet light (it is only 5% but that contribution is important). If this is the case, they should change along all the text “visible light” by sunlight. If, on the contrary, the light was purely visible, the argument on line 262-263 is wrong, since there is not enough energy to excite in ZnO, its bandgap is 3.2 eV, and therefore there are no electrons in the conduction band of the ZnO”
This issue remains unclear. They still assuming saying (lines: 256-258) that there is photoxcitation in the ZnO which can only happen with UV light, unless its bandgap has been modified. But that modification does not seem the case since in their figure 6 the samples effectively absorb in the UV. Have they calculated the bandgap? They should give those values. So it seems UV light contribution is doing something. The text should be changed according.
Author Response
Dr. Viorica-Elena Podasca
Romanian Academy
“Petru Poni” Institute of Macromolecular Chemistry
41 A Gr. Ghica Voda Alley, 700487 Iasi-Romania
March 09, 2020
Answers to Reviewer 1:
Dear Professor,
Thank you for your effort in reviewing our paper entitled “Photopolymerized Films with ZnO and Doped ZnO Particles Used as Efficient Photocatalysts in Malachite Green Dye Decomposition”, authors Viorica-Elena Podasca and Mariana-Dana Damaceanu (manuscript ID: applsci-727140). After a deep examination of the manuscript and taking into consideration the reviewer’s comments, it was accordingly revised, as follows (in red color).
Comment 1: In lines 107-108 they said “Thus, due to the overlap of the peaks, it can be assumed that the ZnO-SnO2 sample is in an amorphous phase [50-52].” But it is no due to overlap of peak, is due to all the reason exposed in 1st revision. Overlapping peaks can make difficult the assignment, but it doesn’t justify amorphous. Rewrite that discussion
Reply 1: We modified the text with: “Thus, it can be assumed that the ZnO-SnO2 sample is in an amorphous phase [50-52].”
Comment 2: In those lines they say Sn oxide is good in several applications. And that’s all. But if their aim was to “dope” (what they didn’t get) they should explain what they expect from doping: structural modifications, electronic changes, bandgap modification etc
Reply 2: The text in lines 73-74: “the purpose of this doping was to make structural modifications and the possibility to use these particles in visible light photodegradation experiments.”
Comment 3: Xe lamp is known commercially and widely in the literature as a solar light simulator. Therefore, they may not use the term sunlight and use "simulated solar ligh" instead
It is not necessary to go to the specifications of the brand of your lamp. It is known that the Xe spectrum has a UV contribution. In addition, the contribution of 5% UV is not negligible and can also intervene in the catalytic process.
And this is also important for the next point:
Comment in 1st revision: “Line 351. Indeed, a Xe-lamp is a solar light simulator. The doubt is: did they use a bandpass filter to select the 437 nm wavelength? If they did not use a filter and the light emitted by the lamp is irradiated directly, the light is not pure visible, it has a contribution of ultraviolet light (it is only 5% but that contribution is important). If this is the case, they should change along all the text “visible light” by sunlight. If, on the contrary, the light was purely visible, the argument on line 262-263 is wrong, since there is not enough energy to excite in ZnO, its bandgap is 3.2 eV, and therefore there are no electrons in the conduction band of the ZnO”
This issue remains unclear. They still assuming saying (lines: 256-258) that there is photoxcitation in the ZnO which can only happen with UV light, unless its bandgap has been modified. But that modification does not seem the case since in their figure 6 the samples effectively absorb in the UV. Have they calculated the bandgap? They should give those values. So it seems UV light contribution is doing something. The text should be changed according.
Reply 3: We modified the term “visible light” with "simulated solar light" as requested.
As was mentioned before we did not used a bandpass filter, the visible light source Xe lamp, λ = 400–800 nm, and its maximum is at 437 nm, this maximum value was taken from the lamp description documents that we received when we purchased the lamps.
The band gap for ZnO particles is 3.3 eV, while for the ZnO-SnO2 particles is 2.9 eV, these values were calculated for the particles only and hence we conducted the experiments under simulated solar light. The values were calculated according to literature [Guy N.; Ozacar M. The influence of noble metals on photocatalytic activity of ZnO for Congo red degradation. Int. J. Hydrog. Energie 2016, 41, 20100-20112; Liu T.; Chen W.; Hua Y.; Liu X. Au/ZnO nanoarchitectures with Au as both supporter and antenna of visible-light. Appl. Surf. Sci. 2017, 392, 616-623]
Since the arguments from lines 262 do not seem right to you and taking into account your requirements, we have shortened this paragraph.
With hope that most of the problems raised from the reviewer were solved, we are waiting for your opinion.
Thank you for your assistance.
Dr. Viorica-Elena Podasca, corresponding author
